# Antimicrobial Locks in Patients Receiving Home Parenteral Nutrition

**DOI:** 10.3390/nu12020439

**Published:** 2020-02-10

**Authors:** Dane Christina Daoud, Geert Wanten, Francisca Joly

**Affiliations:** 1Department of Medicine, Division of Gastroenterology, Centre Hospitalier de l’Universite de Montreal (CHUM), Centre de Recherche du Centre Hospitalier de l’Universite de Montreal (CRCHUM), 1051 Rue Sanguinet, Montreal, Québec, H2X 3E4, Canada; 2Intestinal Failure Unit, Department of Gastroenterology and Hepatology, Radboud University Nijmegen Medical Centre, Geert Grooteplein Zuid 10, 6525 GA, Nijmegen, The Netherlands; geert.wanten@radboudumc.nl; 3Center for Intestinal Failure, Department of Gastroenterology and Nutritional Support, Reference Centre of Rare Disease MarDI, Hopital Beaujon, University of Paris Inserm UMR 1149, 92110 Clichy, France; francisca.joly@bjn.aphp.fr

**Keywords:** catheter-related bloodstream infection, antimicrobial lock therapy, home parenteral nutrition, biofilm, catheter salvage

## Abstract

Catheter-related bloodstream infection (CRBSI) is one of the most common and potentially fatal complications in patients receiving home parenteral nutrition (HPN). In order to prevent permanent venous access loss, catheter locking with an antimicrobial solution has received significant interest and is often a favored approach as part of the treatment of CRBSI, but mainly for its prevention. Several agents have been used for treating and preventing CRBSI, for instance antibiotics, antiseptics (ethanol, taurolidine) and, historically, anticoagulants such as heparin. Nonetheless, current guidelines do not provide clear guidance on the use of catheter locks. Therefore, this review aims to provide a better understanding of the current use of antimicrobial locking in patients on HPN as well as reviewing the available data on novel compounds. Despite the fact that our current knowledge on catheter locking is still hampered by several gaps, taurolidine and ethanol solutions seem promising for prevention and potentially, but not proven, treatment of CRBSI. Additional studies are warranted to further characterize the efficacy and safety of these agents.

## 1. Introduction

Chronic intestinal failure (CIF) is defined as “the reduction of gut function below the minimum necessary for the absorption of macronutrients and/or water and electrolytes, such that intravenous supplementation is required to maintain health and/or growth” [1]. CIF may be caused by different gastrointestinal and systemic diseases with various mechanisms such as short bowel syndrome (e.g., mesenteric ischemia), motility disorders (e.g., scleroderma), intestinal fistulas and extensive parenchymal disease (e.g., Crohn’s disease, Coeliac disease). Thereby, long-term home parenteral nutrition (HPN), mostly administered through a central venous catheter (CVC), has become a life-preserving treatment for these patients. Unfortunately, CIF remains associated with significant morbidity and mortality. For instance, catheter-related bloodstream infections (CRBSIs) are a common complication of long-term HPN and often lead to costly hospital re-admissions [2]. Moreover, repeated CRBSIs may lead to failed venous access, which is an indication for intestinal transplantation [1,3]. Therefore, besides adequate patient and caregiver training to handle the venous access device, additional preventive measures and, if necessary, treating CRBSI is of key importance in the management of patients with CIF.

Currently, central venous catheter-related infections are mostly managed based on protocols that are in accordance with the Infectious Diseases Society of America (IDSA) guidelines by Mermel et al. [4]. Since the release of these guidelines over a decade ago, several studies have suggested that the use of a number of catheter locking agents may further reduce the risk of CRBSI. Moreover, the European Society for Clinical Nutrition and Metabolic Care (ESPEN) has more recently generated several recommendations for the care of adult patients with CIF, including the management of CVC [1]. Since several pivotal studies appeared after the release of these guidelines in 2016, clear recommendations regarding antimicrobial catheter locking in the setting of CIF are so far lacking.

The purpose of this review, therefore, is to provide an overview of current practices and research on the use of antimicrobial catheter locks as preventive and curative measures in adult patients with CIF and to identify current gaps in our knowledge that require future research.

## 2. CRBSIs

According to ESPEN guidelines on CIF in adults, which uses the Centers for Disease Control and Prevention definition, CRBSI is defined by a positive culture of the catheter on removal or paired blood cultures from a peripheral vein and the catheter or by the presence of clinical symptoms of sepsis and the absence of another source of infection. Diagnosis should be achieved by paired quantitative or qualitative blood cultures from a peripheral vein and from the catheter, with continuous monitoring of the differential time to positivity. CRBSI must be distinguished from catheter colonization which is defined as growth of >15 colony-forming units in semi-quantitative culture, or >10^3^ colony-forming units in quantitative culture from a proximal or distal catheter segment, in the absence of clinical symptoms [1,5]. Nevertheless, the definition of CRBSI varies throughout the current literature and is still a matter of debate. Unfortunately, this variability of CRBSI definition has an important impact on our interpretation of the available trials and prevents us from achieving key performance indexes. As such, this disparity in the available studies greatly influences epidemiological data on the incidence of CRBSI, which varies between 0.27 and 11.5 CRBSIs per 1000 catheter days. This is important, since CRBSI rates are considered as a proxy for the quality of HPN care in a centre [6,7,8,9,10,11,12]. Interestingly, and possibly related to the gradual introduction and lesser restricted use of prophylactic antimicrobial catheter locking in many centers, a recent meta-analysis (M-A) on the epidemiology of infectious and non-infectious catheter complications in patients receiving HPN showed a lower rate of CRBSI (0.85/1000 catheter days; 95% CI 0.27–2.64/1000) compared to previous rates reported in Dreesen et al. meta-analysis in 2013 [6,7].

### Biofilm

CRBSIs are mainly thought to originate from contamination of the catheter hub [13]. The organisms colonize the outer luminal surface of the catheter, spread downwards, produce extracellular polymeric products and rapidly form (even <24 h) a multicellular adherent biofilm. The potential of a microorganism to adhere to a catheter depends on catheter surface characteristics, as well as on the host and the microbe [13,14]. Both Gram-positive and Gram-negative bacteria can form biofilms. The most common microbial species encountered in CRBSI in HPN patients are skin derived Gram-positive bacteria (53%) such as coagulase-negative staphylococci (24%) or *Staphylococcus aureus* (10%), followed by Gram-negative bacteria (26%) and fungi, mainly candida species (12%) [6]. In total, 50%–70% of catheter biofilm infections are caused by *Staphylococcus aureus* and *Staphylococcus epidermidis* [15]. Thus, one of the strategies developed to prevent biofilm formation is catheter locking with an antimicrobial solution such as antibiotics, ethanol 70%, taurolidine or anticoagulants. Unfortunately, the biofilm matrix composition protects the bacteria against the immune system as well as these antimicrobial solutions which may explain the loss of efficacy of most locks, leading to recurrent infections [16].

## 3. Curative Treatment of CRBSI

When a CRBSI is suspected, the first question that should arise is whether catheter salvage is possible since repeated catheter removal with loss of suitable vessels eventually compromises the options to obtain adequate venous access. Nonetheless, data concerning successful catheter salvaging in patients receiving HPN remain limited. A recent retrospective study from the Mayo Clinic reported an overall salvage rate of 70% in 1040 patients between 1990 and 2013, with an overall rate of CRBSI of 0.64/1000 catheter days. Furthermore, these authors noticed that salvage rates went from 62% between 1990 and 1994 to 76% from 2010 to 2013. The authors attributed this major improvement to a better multidisciplinary management throughout the years [17]. Few other recent retrospective trials have reported salvage rates ranging between 72% and 91%, with success depending on several factors including causative pathogen and hospital-specific issues related to antibiotic treatment (antimicrobial spectrum, concentration, and duration of treatment) and locking dwell time [9,11]. Current IDSA guidelines recommend immediate removal of the CVC in patients with septic shock, tunnel or port infection, metastatic infection, blood stream infection that continues despite 72 h of antimicrobial therapy and infections due to *Pseudomonas aeruginosa*, mycobacteria or fungi. As for *Staphylococcus aureus*, while the IDSA recommends removal of the catheter, ESPEN guidelines advocate an attempt to salvage the catheter in uncomplicated infection [1,3,4]. When a patient is clinically stable and does not suffer from the above conditions, CVC salvage may be attempted for almost all non-virulent bacterial CRBSI. For instance, ESPEN guidelines advocate salvaging simple infections due to coagulase-negative staphylococci, *Staphylococcus aureus* and Gram-negative bacilli. When only a few options for venous accesses remain, even salvaging fungal CRBSI may be attempted in selected patients. Here, amphotericin B and echinocandin have shown good results with candida infections and seem to be a promising alternative to fluconazole [18].

Salvaging a catheter implies the administration of a systemic antibiotic concomitantly with an antibiotic lock therapy (ALT). The antibiotic lock solution is usually left in the catheter for 12 to 24 h before being withdrawn and instillation of the next lock therapy [4]. Dwell times should generally not exceed 48 h because antibiotic concentrations in the distal lumen of the catheter can rapidly and significantly decrease to subtherapeutic levels [19]. Antibiotic locks have a direct bactericidal activity on the biofilm contrary to systemic antibiotics, which often do not attain therapeutic concentrations in the lumen and, even more so, within the biofilm, which is a major cause for therapy failure and infection recurrences. In order to establish biofilm eradication, it is generally believed that antibiotic concentrations should be 100 to 1000 times above the minimum inhibitory concentration of the causative pathogen [4,19,20,21,22,23]. Since antibiotics lock solutions are often mixed with an anticoagulant such as heparin, it should also be taken into account that catheter locking is not possible for all antibiotics due to precipitation in the catheter at higher concentrations and when combined with heparin [4,24,25]. Currently, the most suitable antibiotics that are used as lock solutions for the treatment of CRBSI with optimal stability include vancomycin, teicoplanin, gentamycin, ciprofloxacin, daptomycin, ceftazidime, cefazolin, minocycline, amikacin and in certain circumstances amphotericin B, echinocandins, and fluconazole [4,18,23,26,27,28,29,30,31,32,33,34]. The desirable concentrations of the most common antibiotic lock solutions used for the treatment of CRBSIs are listed in Table 1.

The duration of ALT that is reported varies widely between studies and depends on numerous factors such as microbial pathogen and patient factors like the presence of prosthetic material such as endovascular implants or orthopedic hardware [4]. As such, salvage duration is often decided empirically on an individual basis and according to local protocols. The IDSA recommends administering ALT for 7 to 14 days which is also in concordance with ESPEN guidelines [1,3,4]. However, further research is needed to better identify the minimum time required to avoid salvage failure and recurrences.

Besides antibiotics, other antimicrobial lock solutions such as ethanol have been proposed as alternatives to ALT [35]. So far, clinical studies with head to head comparisons of the therapeutic effect of these antimicrobial solutions are lacking. A recent in vitro comparison of various catheter locks that are used in the treatment of CRBSIs found that ethanol 70% and taurolidine were the most effective agents. In this study, Visek et al. exposed catheters to different pathogens (*Staphylococcus epidermidis, Staphylococcus aureus*, methicillin-resistant *Staphylococcus aureus*, *Pseudomonas aeruginosa, Candida albicans*), and these were afterwards incubated with a lock solution (ethanol 70%, taurolidine [Tauro-lock Hep_100_], gentamicin, vancomycin). Antibiotics were effective after many hours of treatment whereas the antiseptic (non-antibiotic) compounds (ethanol and taurolidine) eradicated the above pathogens in less than 2 h [36]. Recently, another in vitro study evaluating the effectiveness of a 3 day treatment with heparinized 40% ethanol lock solution showed a decrease in metabolic activity and biomass of biofilms in clinical isolates from patients with CRBSI. However, these authors observed regrowth of the biofilm within 72 h after ending the locking therapy, showing the importance of ongoing prophylactic catheter locking in selected patients [37].

Unfortunately, apart from these in vitro experiments, there are only three controlled clinical trials on therapeutic locking with non-antibiotic antiseptic agents available—two of which come from the pediatric literature [38,39,40]. The single-adult controlled trial so far concerns a phase II study in the US that assessed the efficacy of a combined solution of minocycline-EDTA-ethanol 25% as a lock therapy in adult patients with CRBSI. These authors demonstrated that this lock therapy decreased the rate of mechanical and infectious complications (CRBSI, septic thrombophlebitis, unsuccessful insertion attempts and misplacement of the catheter) when compared to removing the catheter. In addition, patients with this lock intervention required a shorter duration of systemic antibiotic therapy and retained their CVC for more than 2 months after the onset of the CRBSI. However, importantly, this trial focused on patients suffering from malignancy who underwent chemotherapy, a population that differs considerably in numerous aspects from our CIF population receiving HPN [41].

Although the use of thrombolytic locks (urokinase, streptokinase, tissue plasminogen activators) as an adjunctive therapy for CRBSI is not currently recommended by the IDSA and ESPEN guidelines, some centers use them during salvage as it may contribute to the dissolution of the intraluminal fibrin envelope [42].

So far, several aspects of CVC salvage (duration of the attempt, definition of success) and even the definition of CRBSI remain a matter of debate that urgently requires more consensus to facilitate clinical and research practices.

## 4. Prevention of CRBSI

Throughout the years, numerous studies have demonstrated that the cause of CRBSI is multifactorial. Reported risk factors associated with CRBSI are related to the patient (e.g., presence of a stoma, opiate use), the catheter (type, number of lumina), quality of education and adherence to protocols by staff and patients, regimens for parenteral nutrition therapy and follow up [5,7,11,43,44].

Also, because most CRBSI are thought to originate from an infected catheter connection hub and in order to decrease the infection rate, numerous strategies have been developed to prevent intraluminal catheter colonization. As such, the use of a tunneled single lumen catheter that is dedicated for total parenteral nutrition administration and training of patients to rigorously perform aseptic catheter handling procedures currently establish the standard of care in expert centres [1,5]. Importantly, it has been shown that regular replacement of the catheter and the use of in-line filters fail to prevent CRBSI and these techniques should no longer be used [1,3,5].

As for the prophylactic use of catheter locking, various solutions have been studied, including antibiotics, antiseptics (ethanol, taurolidine) and anticoagulants (heparin, EDTA). By instilling an antimicrobial lock solution for several hours while the catheter is not in use, the aim is to prevent bacterial attachment and, hence, biofilm formation. In recent years, the use of several of these agents has been abandoned because of development of microbial resistance (antibiotics), lack of efficacy or even suspected promotion of infections (heparin) or side effects [3,5,45]. For instance, it has been shown that the antimicrobial activity of heparin locks is lost when administered at a concentration below 6000 U/mL and that this agent can even contribute to biofilm formation and *Staphylococcus aureus* growth [5,46]. On the other hand, tetrasodium ethylenediamine tetraacetic acid (EDTA), an alternative to heparin, is being trialed for the prevention of biofilm formation and seems to be a promising antimicrobial compound, as was demonstrated in a recent Canadian in vitro study [47].

### 4.1. Ethanol 70%

In addition to its potency to unclog catheters and dissolve lipid deposits, ethanol 70% has also been found to have both bactericidal and fungicidal properties as well as a high activity against biofilm formation, supporting its use as a preventive measure. Ethanol damages proteins and consequently, unlike antibiotics, lowers the risk for microbial resistance [35,48,49]. Opilla et al. conducted a prospective study using prophylactic ethanol lock therapy (ELT) in 9 patients with recurrent CRBSI. This trial showed a significant drop in CRBSI rates from 8.3 per 1000 (before ELT) to 2.7 per 1000 catheter-day (after ELT, RR 0.325; 95% CI, 0.17–0.64) [50]. Subsequently, a retrospective study using data from the Cleveland Clinic HPN database evaluated CRBSI-related admission rates pre- and post-ELT and found in 31 patients that CRBSI rates were reduced from 10.1 to 2.9 per 1000 catheter days. Also, no complications were reported [51].

Despite the supposed advantages, a systematic review has shown that adverse events remain the Achilles’ heel of ethanol locks including the (concentration-dependent) occurrence of plasma protein precipitation, and increased risk of thrombosis, catheter damage and systemic toxicity [52]. Another recent meta-analysis on the overall use of ethanol lock in prevention of CRBSI, which included 10 randomized controlled trial (RCT) and 2760 patients, demonstrated that ethanol locking significantly diminishes the incidence of CRBSI (RR 0.66, 95% CI 0.51–0.86). Also, subgroup analyses based on differences in study quality and duration of the locking were also in favor of ethanol over the standard of care. In fact, the results of this meta-analysis showed no increase in the incidence of thrombosis (RR 1.05, 95% CI 0.51–2.18). Differences in patient population may play a role here since this latter meta-analysis did not only include HPN patients but considered all studies with long-term CVC use, such as patients on hemodialysis or chemotherapy [53]. Nevertheless, in light of these contradicting results and because of the lack in strong data, ESPEN guidelines recommend against catheter locking with 70% ethanol for the prevention of CRBSI in patients on HPN [4,52].

### 4.2. Taurolidine

Taurolidine, a relative “new kid on the block” in prophylactic catheter locking, is known for its bactericidal activity against a very broad antimicrobial spectrum. This antiseptic agent is a derivate from the amino acid taurine. It inhibits microbial adhesion to biological surfaces by means of an irreversible chemical reaction of the active N-methylol group formed upon breakdown of the molecule, with microbial cell wall components [54]. Several studies comparing taurolidine to various other catheter locking solutions have been conducted to analyse its efficacy as CRBSI prophylaxis, its effect on the biofilm, its cost-effectiveness and the development of microbial resistance. The efficacy of taurolidine as a prophylactic catheter lock in HPN patients was first reported in 1998 in a case report by Jurewitsch B. and subsequently in a non-controlled cohort of 7 patients on HPN [55,56]. In 2009, Bisseling et al. conducted the first randomized open-label controlled trial comparing taurolidine 2% (*N* = 16) to low dose heparin (*N* = 14). Regardless of its low patient number, this study was a pioneer by demonstrating taurolidine’s superiority to heparin. While 10 relapses were reported in the heparin group, only one re-infection was observed in the taurolidine group. Besides, the mean infection-free survival was 176 days in the heparin arm versus 641 days in the taurolidine arm (*p* < 0.0001). After crossing over of patients who developed a CRBSI under heparin, only one re-infection occurred. No side effects or catheter occlusions were reported in both groups [57]. In 2014, Klek S. et al. prospectively conducted a trial comparing taurolidine 2% (*N* = 10) to taurolidine 1.35% + citrate (*N* = 10) or to a saline group (*N* = 10) administered to patients with a low infection rate. After one year of follow-up (total of 10,968 catheter days), only 1 CRBSI was observed in the taurolidine 2% group. Most probably because of the low patient number, this study did not detect any clinical advantage or cost-effectiveness of taurolidine in patients with an a-priori low risk for CRBSI [58]. A recent international multicentre (Denmark, Israel, Italy, The Netherlands, United Kingdom), double-blinded superiority trial compared taurolidine 2% to 0.9% Saline (NS). After being assigned to either taurolidine or NS, patients were also stratified in two distinct groups: the ‘new catheter group’ (new patient on HPN or new catheter) and the ‘pre-existing catheter group’ (HPN > 1 year and catheter for > 6 months) which was considered as the higher risk group because of the possible pre-existing biofilm. As expected, this study showed that taurolidine significantly reduced the risk for CRBSI by more than four time. This difference was mainly present in the new catheter group where the rates of CRBSI/1000 catheter days were 0.29 in the taurolidine arm and 1.49 in the saline arm (RR 0.20; 95% CI, 0.04–0.71; *p* = 0.009). In the pre-existing catheter group, the difference between CRBSI rates was, however, not statistically significant. The authors suggest that these results may be due to a limited number of patients as well as to a pre-existing biofilm which is known to be one of the most important precursors of CRBSI. Moreover, according to this trial, an additional cost effect benefit using taurolidine was demonstrated. The mean costs per patient was 1865$ (95% CI, $1016–2931) for taurolidine compared to 4454$ (95% CI, $2631–6579) for saline (*p* = 0.03) [59]. In 2013, Liu et al. published a meta-analysis on taurolidine lock as a prevention for CRBSI. 6 RCT conducted between 2004 and 2013 were included in this review with a total of 431 patients and 86,078 catheter days. This meta-analysis demonstrated that when compared to heparin, taurolidine did reduce the incidence of CRBSI [60]. Moreover, in order to compare the antimicrobial activity of various taurolidine catheter lock formulation whether with different concentration or in combination with an anticoagulant, Olthof et al. conducted an in vitro trial where they collected clinical isolates obtained during CRBSI on HPN patients and grew them in various taurolidine formulations: taurolidine 2%, 1.34% taurolidine + citrate, 1.34% taurolidine + citrate + heparin or phosphate buffered saline (control). The analysis of biofilm formation and growth of clinical isolates demonstrated that all taurolidine containing lock solutions, regardless of the presence of an anticoagulant, prevented microbial growth. A significant difference in microbial growth was found between diluted taurolidine 2% and 1.34% formulations—the clinical significance of which remains to be proven. *Escherichia coli, Staphylococcus aureus* and *Candida glabrata* growth were inhibited 10 h longer with taurolidine 2% compared to the 1.34% concentration [61].

Until now, fungal infections remain daunting because of the high associated rate of morbidity and mortality and because this type of infection requires immediate removal of the catheter. In order to evaluate which lock solution is most potent to eradicate biofilms colonized by fungi, Rosenblatt et al. designed an in vitro study comparing saline, heparin, citrate, taurolidine + citrate + heparin and nitroglycerin + citrate + ethanol (NiCE). Then, 60 min after the exposure to the lock solution, the NiCE solution was the only one to fully eradicate the candida biofilms [62]. Even though these pre-clinical results are promising, further studies are required to assess if ethanol lock solutions can diminish the rate of fungal CRBSI and whether the beneficial effects outweigh the previously discussed adverse effects of ethanol. Studies on taurolidine and ethanol lock for the prevention of CRBSI in patients receiving HPN are listed in Table 2.

### 4.3. Adverse Effects of Long-Term Use of Antimicrobial Lock Solutions

The quality of any antimicrobial lock is not only defined by its potency to decrease infection rates but also by its spectrum of adverse effects. Long-term catheter locking may be associated with certain risks. For instance, prophylactic use of antibiotics has been abandoned because of microbial resistance [3,45]. Also, evidence suggests that low concentrations of heparin promote biofilm formation [46]. As for ethanol lock and taurolidine, significant differences in terms of safety have been reported. While ethanol locking has been associated with the development of catheter damage and systemic adverse effects such as disseminated intravascular coagulation, deep venous thrombosis and hemolysis, the risk for side effects related to taurolidine seems relatively low [58]. A recent retrospective study describing the long-term clinical outcomes of 270 HPN patients that used taurolidine for 338,521 catheter days, reported low central venous access device (CVAD) related complications rates as well as mild to moderate adverse events such as pain related to the instillation of the solution, nausea and pruritus. The rates of CVAD thrombosis and occlusion were 0.28 (CI 95% 0.23–0.34), and 0.12 (CI 95% 0.08–0.16) events per 1000 catheter days respectively [64]. Moreover, in addition to their broad spectrum of antimicrobial activity, there is no evidence for microbial resistance with neither taurolidine or ethanol lock, which is not surprising given their mode of action which significantly differs from that of antibiotics [36,52,65]. Table 3 list the adverse effects associated with each antimicrobial lock.

## 5. Conclusions

Since the threat of CRBSI is one of the foremost burdens for HPN patients, prophylactic catheter locking with an antimicrobial solution is becoming one of the hot topics in this field that can extend the life span of CVC and improve the patient’s quality of life. Nevertheless, it should be kept in mind that the use of any lock should never replace rigorous training and strict adherence to aseptic catheter handling protocols.

Many questions still need to be answered to characterize the optimal lock formulation and duration of treatment to establish catheter salvage when feasible and without jeopardizing the patient because of an undertreated infection. This notion also, and even more so, applies to fungal infections. Given the wide variety of available protocols, prospective trials addressing frequency and dwell time of prophylactic antimicrobial locks are also needed to further define and confirm their role in preventing CRBSI in HPN patients. Since the number of well-powered studies on antimicrobial locks in the setting of HPN remains low so far, with heterogenous design, and limited power, and in the absence of head-to-head comparisons of currently available locks, the ideal catheter lock solution at this point remains elusive [6].

While the use of antimicrobial lock therapy for salvaging is an approach that more or less has become part of usual HPN care, prophylactic locking sometimes remains a more controversial subject in terms of what agent to use and in which patient to consider it. Indeed, the question of whether prophylactic antimicrobial locks should be used as a primary prevention strategy (low risk patients) or should be restricted to patients with increased risk for CRBSI remains unanswered.

Furthermore, catheter locking may be considered a double-edged sword therapy as it decreases the rate of CRBSI, but its long-term use might be associated with non-negligible complications such as catheter damage with ethanol lock therapy, or microbial resistance in case of antibiotics. For the moment, taurolidine seems to be the best option in terms of its low associated complication risk.

In conclusion, inhibiting microbial adherence to the catheter and preventing the formation of a biofilm remains key in the prevention of CRBSI. Since penetration of antimicrobial lock solutions into the biofilm is limited, novel avenues of antibiofilm technology should be explored. A recent systematic review on such strategies for the prevention and treatment of biofilm-related infections has showed several promising devices, such as catheters with a modified submicron surface texture or with polymer brush coatings. Only the future will show whether and how this exciting new technology will become part of our therapeutic clinical armamentarium [66].

## Figures and Tables

**Table 1 nutrients-12-00439-t001:** Concentration of common antibiotic lock solutions used for the treatment of catheter-related bloodstream infection (CRBSI).

Antibiotic	Antibiotic Concentration	Heparin Concentration	Reference
With anticoagulant			
Vancomycin	5 mg/mL	5000 units /mL	Lee et al., 2006 [31]
Gentamycin	1 mg/mL	2500 units/mL	Krishnasami et al., 2002 [30]
Ceftazidime	0.5 mg/mL	100 units/mL	Rijinders et al., 2005 [33]
Cefazolin	5 mg/mL	5000 units/mL	Krishnasami et al., 2002 [30] Vercaigne et al., 2000 [34]
Without anticoagulant			
Teicoplanin	10 mg/mL	-	Lee et al., 2006 [31]
Amikacin	1 mg/mL	-	Lee et al., 2007 [32]
Ciprofloxacin	5 mg/mL	-	Lee et al., 2006 [31] Lee et al., 2007 [32]

**Table 2 nutrients-12-00439-t002:** Studies on taurolidine and ethanol lock for the prevention of CRBSI in adults receiving home parenteral nutrition.

Author, Year	Study Design	Antimicrobial Solutions	Patients (N)	Study Years	Follow-Up Duration	Results
Bisseling et al., 2010 [57]	RCT	TL 2% vs. HL	30	2006–2008	HL: 353 ± 51 days TL: 336 ± 51 days (mean)	HL: 2.02 (1.1–3.8) CRBSI/1000 days TL: 0.19 (0.003–1.3) CRBSI/1000 days (*p* = 0.008)
Liu et al., 2013 [60]	M-A	TL 2% vs. HL	431	2004–2013	In total, 31 to 349 days	RR of CRBSI: 0.34 (0.21–0.55; *p* < 0.0001)
Klek et al., 2015 ^a^ [58]	RCT	TL 2% vs. TL 1.35% + citrate vs. saline	30	2012–2013	In total, 12 months	TL 2%: 0 CRBSI / 1000 days TL 1.35% + citrate: 0.273 CRBSI / 1000 days Saline: 0 CRBSI / 1000 days (*p* = 1.00)
Wouters et al., 2018 [59]	RCT	TL 2% vs. saline 0.9%	85	2013–2015	TL: 363 (119–370) days/patient Saline: 346 (109–368) days/patient (median)	TL: 0.33 (0.11–0.76) CRBSI/1000 days Saline: 1.44 (0.85–2.23) CRBSI/1000 days Relative risk: 0.23 (0.07–0.63) (*p* = 0.002)
Tribler et al., 2017 [63]	RCT	TL 1.35% + citrate + heparin vs. HL	41	2013–2014	TL: 592 (10–756) days/patient HL: 207 (25–755) days/patient	TL: 0 CRBSI/1000 days HL: 1 (0.4–2.07) CRBSI/1000 days (*p* = 0.0052)
Reitzel et al., 2019 [6]	M-A	TL^b^ vs. EL 70%	713	2012–2019	-	Pre-TL: 0–6.58 CLABSI/1000 days TL: 0–1.1 CLABSI/1000 days Pre-EL: 0.32–12.7 CLABSI/1000 days. EL: 0.47–2.4 CLABSI/1000 days
John et al., 2012 [51]	Retrospective cohort	EL	31	2006–2009	Total of 34 411 catheter days 27 210 catheter days before EL 7201 catheter days after EL introduction	Pre-EL: 3.53 CRBSI/1000 days Post-EL: 1.65 CRBSI/1000 days (*p*= 0.011)
Zhang et al., 2019 [53]	M-A	EL vs. HL	615	2008–2017	-	OR: 0.53 (0.34–0.82; *p* = 0.004)

RCT: randomized controlled trial; M-A: Meta-analysis; TL: taurolidine lock; EL: ethanol lock; HL: heparin lock; RR: risk ratio; OR: odds ratio, CRBSI: catheter-related bloodstream infection, CLABSI: central line-associated bloodstream infection. ^a^ The aim of this study was to analyze the clinical value of taurolidine in patients receiving home parenteral nutrition with a low infection rate. ^b^ Concentrations of taurolidine and/or citrate varied between studies.

**Table 3 nutrients-12-00439-t003:** Adverse effects associated with long-term administration of antimicrobial lock solutions.

Antimicrobial Lock Solution	Adverse Events	References
Antibiotics	Bacterial resistance	Pittiruti et al. 2009 [3] Mermel et al. 2009 [4]
Heparin ^a^	Promotion of *Staphylococcus aureus* growth and biofilm formation	Shanks et al. 2005 [46]
Taurolidine ^b^	No serious side effects were reported	Wouters et al. 2018 [59] Wouters et al. 2019 [64] Klek et al. 2015 [58] Bisseling et al. 2010 [57] Olthof et al. 2013 [65]
Ethanol	Catheter damage Systemic toxicities (DIC, deep venous thrombosis, hemolysis) Alcohol taste, nausea, vomiting	Mermel et al. 2014 [52] Visek et al. 2019 [36]

DIC: disseminated intravascular coagulation. ^a^ When heparin is administered at a concentration below 6000 units/mL. ^b^ No serious side effects were reported with different concentrations of Taurolidine (2%; 1.35%).

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
