# Peer review of "Antimicrobial Locks in Patients Receiving Home Parenteral Nutrition"

_nutrients, 2020, doi:10.3390/nu12020439_

Round 1

Reviewer 1 Report

This review article by Dane Christina Daoud and co-workers addressed the antimicrobial catheter locks that currently available to prevent the infections related to chronic intestinal failure. I think the manuscript is generally written well and logically conceived. Besides, objectives are well defined. This article is of great significance for preventive and curative measures. Moreover, the conclusion reflects the main findings, therapeutic options and future research are needed. However, the manuscript needs to be further revised and improved.

The specific comments are as follows:

Please put the line number for comfortably reviewing the manuscript by reviewers.

Abstract

Sentence “several agents have been used”… please specify for what

Sentence “seem promising approaches” replace the word approaches

 Sentence “CIF may be secondary … the main mechanism, not clear. Please rephrase.

Paragraph CRBSIs

 Sentence “colony-forming units” is there specific bacteria to test on blood culture. If do so, please specify.

Paragraph Biofilm

Coagulase-positive Staphylococcus Aureus change to Staphylococcus aureus

Sentence “Both Gram-positive and Gram-negative bacteria can form biofilms”. Move to the beginning

Paragraph Curative treatment of CRBSI

Pseudomonas Aeruginosa change to Pseudomonas aeruginosa

 In addition to antibiotics available to treat these infections, is there any report on the use of carbapenem group of antibiotics?

Pseudomonas Aeruginosa, Candida Albicans… Italicize

Conclusion

 Have the authors investigated the age and other disease effects while using these treatments?

Reviewer 2 Report

This paper is a review article addressing the role of antimicrobial locks in patients receiving home parenteral nutrition. It describes existing related literature and discusses common current practices as well as identifying areas of paucity in the literature / role for future research / unanswered questions. It is predominantly focussed on adult patients given much of related current literature comes from this population. 

It is a well written paper that not only outlines the literature in detail but provides good discussion and synthesis around this in a manner that is applicable to clinicians for use in their clinical practice. Limitations of the existing literature are addressed in various different sections rather than as an isolated section.

I have a few minor English corrections and suggestions:

Section 2. CRBSIs

Line 10-11. This is a very important point about differing definitions used in the literature for CRBSIs and some further discussion could be given to the implications of this on comparing between studies and interpreting literature for the purpose of generating clinical protocols, achieving key performance indexes etc.

Section 2.1 Biofilm.

'A microorganism potential to adhere...'- suggest change structure of sentence e.g. 'The potential of a microorganism to adhere...'

Section 3. Curative treatment of CRBSI

Line 1: 'the first question that should arises' - need to remove 'should' or change arises to arise

p 5 of 14: 'Canadian' instead of 'canadian'

Table 3: foot note (a) regarding Heparin - concentration requires volume to be stated e.g. 6000 units / mL.
